# OpenReview forum: "Reinforcement Unlearning via Group Relative Policy Optimization"
_ICLR.cc/2026/Conference — ICLR 2026 Poster_

### Official Review · Reviewer_Kspv · 2025-10-30

**Soundness:** 2
**Presentation:** 3
**Contribution:** 3
**Rating:** 6
**Confidence:** 3

**Summary:**

The paper proposes PURGE, an unlearning RL framework for LLM. The method is built on the idea of Group Relative Policy Optimization. PURGE frames unlearning as a verifiable task with an intrinsic reward that penalizes output containing any tokens from a specified forget vocabular. It constructs this vocabulary by probing the base model with RWKU’s rejection-tuning questions and then running a conditioned NER to produce a set of terms; training then maximizes group-relative advantages with a KL regularizer to preserve the model performance. The method includes theoretical results such as leakage rates for unlearning guarantee and a utility bound that relates downstream performance to the KL-divergence. The proposed framework is evaluated on the RWKU benchmark across an array of metrics. Gains over baselines are competitive or stronger than SOTA on several metrics.

**Strengths:**

1. The paper is generally well written with self-contained and consistent use of notation. The idea is also clear and well-motivated, casting unlearning as a verifiable objective with a simple indicator reward, no expensive external reward model. The GRPO objective and vocabulary construction are described concretely.

2. The framework feels pipelined and easy to use, the idea of using probing to elicit model-specific answers and NER to produce forget target, and is practical and general, applicable to many unlearning methods.

3. Theoretical guarantees match the intended contribution, providing insights on both the forgetting capability (Theorem 1) and relating downstream performance with the KL-divergence, which can be explicitly controlled during the unlearning procedure (Theorem 2).

4. Solid empirical studies are provided using RWKU. The main table shows improved forgetting vs. base and competitive SOTA methods. The main PURGE benefits (efficiency per target) are also clearly shown.

5. Ablations study shows robustness against 8/9 adversarial attacks, and scaling curves across Qwen sizes demonstrating monotonic improvements in forgetting.

**Weaknesses:**

1. Keyword or vocab-based unlearning does not consider obfuscation and homonym issues. This suggests surface suppression rather than semantic forgetting.

2. Surface-level keyword suppression does not entirely solve the issue of memorizing sensitive information. There is a gap between the real intention of machine learning (to produce a model as if the model never sees targeted information) and the contribution of this paper (to suppress keywords associated with that information).

3. The proposed method heavily relies on an additional large language model for forget corpus construction. However, there isn't a way to directly assess the quality of the targeted corpus itself or how well it is associated with the actual unlearning target.

4. The mixing rate alpha does not seem to be reflected in the algorithm itself, but is the key assumption for Theorem 1 on the suppression guarantee.

5. The scope and evaluation presented in this paper are somewhat limited, mainly focusing on RWKU. Its generalizability broader privacy/safety settings is deferred to future work.

**Questions:**

1. Theorem 1 assumes mixing of policies instead of a fixed policy. Please clarify whether the mixing rate α is used and how it is reflected in the algorithm. If not, how does it affect the guarantee?

2. How or could the proposed unlearning mechanism be transferred beyond keyword-forgetting? Do authors have a semantic or entailment-based criterion to prevent typos or obfuscation?

3. Why is GRPO preferable for unlearning tasks? Are there any alternatives to GRPO?

---

> ### Author Response · Authors · 2025-11-20
> **Part 1**
>
> We thank the reviewer for the careful review and for highlighting both the strengths of our work and the areas where clarification is needed. We address the questions and weaknesses one by one below.
>
> ### W1 (keyword-based forgetting and robustness to obfuscation):
> We agree that pure surface-form suppression is inherently limited if used in isolation. However, in practice, minimizing the probability of generating characteristic tokens associated with a concept induces broader semantic suppression. Because PURGE penalizes entire completions (i.e., full reasoning trajectories), the model learns to reduce not only explicit mentions but also the contextual semantics in which those mentions typically arise. For example, once the probability of producing “Stephen King” decreases, semantically related tokens such as “The Shining” or “horror novelist” are also suppressed. Our experiments validate that this effect is sufficiently strong to withstand paraphrasing, obfuscation variations, and other common adversarial attacks (see Fig. 4 for details).
>
> ### W2 (surface-form suppression vs. true unlearning):
> We agree this distinction deserves explicit discussion. In current LLM unlearning literature, approaches generally fall into two meta-categories:
> * **Data-point deletion**, where the goal is to remove specific passages or personal information (e.g., copyrighted text), typically addressed with gradient-based deletion or relabeling; and
>
> * **Concept-level suppression**, where the aim is to eliminate the model’s ability to express or reason about certain knowledge domains, usually approached via preference-based RL (e.g., RT, DPO/NPO, Quark).
>
> PURGE is best suited for the latter setting but is not restricted to named entities or keywords. Our verifiable reward can be defined over any forbidden string set, including exact spans, paragraphs, or user-specified canonical representations of protected content. In practice, practitioners can structure forbidden material hierarchically, from coarse to fine, to trade off conceptual robustness against precision. These two categories represent different aspects of the same concept of unlearning, offering distinct perspectives on the problem of unlearning, as noted in [1] and [2].
>
> ### W3 (reliance on an external LLM for forget-corpus construction):}
> Many SoTA methods rely on external LLMs to synthesize large corpora, either for preprocessing/data generation [7,8] or for evaluation [9]. PURGE uses an external model to generate concise descriptors of the target concept. Moreover, the forget vocabulary is conditioned on the target model’s own responses to RWKU probing questions. This ensures that the extracted phrases reflect the model’s internal knowledge, rather than generic world knowledge, and stay semantically close to the unlearning target. Finally, we manually validate the produced descriptors to ensure alignment with the true unlearning target. We include additional explanations and prompt templates in Appendix E.
>
> ### W4 and Q1 (mixing rate $\alpha$):
> For some weird reason, * in the subscript doesn't work. so we use $u$ = *
>
> Thank you for pointing this out. We want to clarify that Theorem 1 analyzes an idealized variant of our training procedure that includes an explicit mixing rate $\alpha$ between the learned policy $\pi_{\theta_t}$ and a fixed base policy $\pi_{\theta^u}$. This mixing mechanism ensures that every update includes a baseline probability of sampling from $\pi_{\theta^u}$, which defines the leakage floor $p_{\theta^u}$. The algorithm described in the main text corresponds to the special case $\alpha = 0$, i.e., no explicit mixture sampling is performed. In this case, the one–step bound simplifies to $p_{t+1} \leq (1 - \eta \varepsilon) p_t,$
> which predicts complete suppression in the limit. More specifically, GRPO is a special case of Theorem 1 where $\alpha = 0$. Theoretically, after $T \to \infty$ iterations, we would reach a probability of 0 for the forbidden tokens since $\lim_{T\to\infty} (1-\eta\varepsilon)^Tp_0 = 0$. In practice, however, various sources of stochasticity (sampling noise, optimization noise, imperfect KL penalties) act as an implicit mixing mechanism. As a result, we observe a nonzero leakage floor in experiments even though $\alpha = 0$ in the implementation. In the revised version, we will make the following points explicit: (1) Theorem 1 analyzes a generalized update rule that includes an
> explicit mixing rate $\alpha$ between $\pi_{\theta_t}$ and a fixed reference policy $\pi_{\theta^u}$; (2) The implemented algorithm corresponds to the special case $\alpha = 0$; (3) In practice, optimization noise induces an effective mixing rate $\alpha > 0$, which explains why the leakage floor predicted by $p_{\theta^u}$ qualitatively matches the empirical plateau (see Figs. 2 and 5).

---

> ### Author Response · Authors · 2025-11-20
> **Part 2**
>
> ### W5 (more benchmarks)
> We focused on RWKU because it jointly evaluates forgetting, utility retention, adversarial robustness, and membership privacy in a unified framework. However, as suggested, we extended the evaluation of PURGE using the **TOFU** benchmark and will include these results in the revision. Only minor adjustments to the forget-corpus construction were required, demonstrating that PURGE generalizes beyond RWKU without relying on RWKU-specific assumptions. We reproduce and compare against other SoTA methods using the OpenUnlearning [10] framework on Llama-3.2-1B-Instruct (as reported in the table in `open-unlearning/docs/repro.md`). Here are the results:
>
> ### TOFU Benchmark Results (**Bold** is best, `Emphasized` is second-best)
>
> | **Method** | **Forget Quality** | **Forget Truth Ratio** | **Model Utility** |
> | ---------- | ------------------ | ---------------------- | ----------------- |
> | Finetuned  | 1.88×10⁻²²         | 0.4753                 | 0.5992            |
> | Retain     | 1                  | 0.6272                 | 0.5909            |
> | AltPO  | **1.46×10⁻⁶**      | **0.6517**             | 0.5715            |
> | GradDiff   | 5.63×10⁻²⁰         | 0.4568                 | 0.5868            |
> | IdkNLL     | 1.49×10⁻¹⁶         | 0.5149                 | 0.5560            |
> | NPO    | `1.62×10⁻¹⁰`       | `0.5378`              | 0.5964            |
> | UNDIAL     | 1.88×10⁻²²         | 0.4805                 | **0.6016**        |
> | RMU        | 6.92×10⁻²¹         | 0.4668                 | 0.5115            |
> | SimNPO     | `1.62×10⁻¹⁰`     | 0.5042                 | 0.5931            |
> | PURGE  | 1.12×10⁻¹⁹         | 0.4843                 | `0.5990`          |
>
> As you can see, PURGE remains comparable with other SoTA approaches (such as GradDiff, RMU, UNDIAL), though it is not the best. This is expected, considering it follows a conceptually different approach than other methods. There is still substantial room for improvements and open questions, which could spark opportunities for future research.
>
> ### Q2 (extend PURGE beyond keyword-based forgetting):
> We refer the reviewer to W2. In addition, we experimented with extracting embedding-based semantic neighbors for each forbidden phrase, but this approach frequently surfaced tokens with high geometric similarity but low semantic specificity (e.g., “Stephen King” → “Steven”, “Alexander”). Identifying genuine semantic entailment relations from embeddings would require deeper mechanistic interpretability, which is beyond the scope of this work. Using an external LLM for controlled concept description was therefore a practical and reliable middle ground between fully manual specification and full interpretability-based discovery.
>
> ### Q3 (Why GRPO for Unlearning? Alternatives?):
> We adopt GRPO primarily for its algorithmic suitability to the unlearning setting. GRPO enables the model to learn through self-exploration how to generate safe, non-leaking responses, rather than relying on fixed refusal templates or manually crafted heuristics. This reduces human-imposed biases in reward specification, allowing the model to autonomously discover robust strategies for suppressing forbidden content. As stated in W4 and Q1, GRPO is a suitable approach since, theoretically, at the limit of $T \to \infty$, the probability of emitting forbidden tokens tends to 0, making it a suitable approach for unlearning. We will clarify this better in the main text.
>
> GRPO is also representative of a broader family of modern RL methods that could, in principle, be used for unlearning. Potential alternatives include REINFORCE++ [3], DAPO [4], Dr.GRPO [5], and BNPO [6]. Since GRPO has not been applied to unlearning tasks in prior literature, we believe that our work provides an innovative solution.
>
> [1] Rethinking Machine Unlearning for Large Language Models, Liu et al.
>
> [2] Bridging the Gap Between Preference Alignment and Machine Unlearning, Feng et al.
>
> [3] REINFORCE++: Stabilizing Critic-Free Policy Optimization with Global Advantage Normalization, Hu et al.
>
> [4] DAPO: An Open-Source LLM Reinforcement Learning System at Scale, Yu et al.
>
> [5] Understanding R1-Zero-Like Training: A Critical Perspective, Liu et al.
>
> [6] BNPO: Beta Normalization Policy Optimization, Xiao et al.
>
> [7] RWKU: Benchmarking Real-World Knowledge Unlearning for Large Language Models, Jin et al.
>
> [8] TOFU: A Task of Fictitious Unlearning for LLMs. Maini et al.
>
> [9] Unlearning That Lasts: Utility-Preserving, Robust, and Almost Irreversible Forgetting in LLMs, Singh et al.
>
> [10] OpenUnlearning: Accelerating LLM Unlearning via Unified Benchmarking of Methods and Metrics, Dorna et al.

---

> > ### Comment · Reviewer_Kspv · 2025-11-27
> >
> > Thanks for the additional experiments; additional results are much appreciated, and I agree that panelizing key terms in reasoning trajectories can, to some degree, be seen as a semantic level removal. I believe there may be other better ways of achieving this goal, but I also see many other works facing the same issue. I also notice that PURGE did not achieve the best (or second best) in many categories. Would you describe the token efficiency as the main strength here (Figure 1 in the main paper)? I also feel that how this efficiency is achieved with regard to other methods (GA, DPO, NPO RT)  is not sufficiently discussed. Can the authors provide more context?
> >
> > I've adjusted my ratings.

---

> > > ### Author Response · Authors · 2025-11-27
> > >
> > > Thank you for the thoughtful comments and for adjusting your ratings. We greatly appreciate it.
> > >
> > > To answer your question, yes, token efficiency is indeed one of PURGE’s primary strengths. This is precisely why we report these results alongside the main performance comparisons. For additional context, methods such as GA, DPO, and NPO require complete, multi-sentence passages describing the target concept from the model's training data. These passages are substantially longer and more detailed than the concise concept descriptors used by PURGE. RT comes closer in terms of efficiency, but PURGE remains more efficient while achieving stronger unlearning performance.
> > >
> > > In our main experiments, we allocate approximately the same total token budget as NPO/GA (with DPO requiring an even larger budget, making it inherently less efficient) to ensure a fair comparison. One might propose to limit the budget of NPO/GA to that of PURGE, but the gap in token volume is so large that one cannot construct a valid forget corpus for GA/NPO using the same limited token budget; there is not enough text to form a meaningful sample to unlearn in their framework. This makes the efficiency improvement attributable specifically to PURGE’s lightweight forget corpus rather than differences in total optimization steps. Overall, PURGE provides significantly higher efficiency per unit of forget information. This makes it particularly advantageous in settings where only limited information about the target concept is available or where generating long concept-specific passages is impractical due to the cost of generation. For clarity and reproducibility, we will expand the discussion in the revision and include a section in the appendix discussing the precise token comparison methodology underlying Figure 1.

---

> ### Author Response · Authors · 2025-11-28
> **This reviewer raised their scores way before the leak**
>
> I am just letting the AC know that this reviewer raised their scores from 6 (original) to **8** (new) at **10:28 CET** November 27th (well before the leak).

---

> > ### Comment · Area_Chair_dZQX · 2025-11-28
> >
> > I think you have received the urgent email where it clearly said "Due to uncertainty around when the exploit was introduced, we are also reverting all reviews and their scores to their state before the start of the discussion period." In fact, the security issue was identified as early as Nov 12 and this is why "before the start of the discussion period".
> >
> > All submissions will be reassigned to new ACs soon, and you are not allowed to post similar comments to your new AC. The AC would make a recommendation with and only with your submission, the initial reviews, and your rebuttal. Thank you for your cooperation.
> >
> > Your old AC

---

### Official Review · Reviewer_FNKG · 2025-10-31

**Soundness:** 3
**Presentation:** 3
**Contribution:** 2
**Rating:** 6
**Confidence:** 3

**Summary:**

The paper introduces a reinforcement learning approach to achieve verifiable and efficient unlearning in large language models. It treats unlearning as a policy optimization problem that reduces the likelihood of generating forbidden content while preserving the model’s overall capabilities. PURGE employs a group-relative policy objective with a KL regularizer to ensure theoretical guarantees such as bounded utility loss and provable forgetting efficiency. Experiments on the RWKU benchmark demonstrate that PURGE performs unlearning much faster, using up to 46 times fewer tokens than existing methods, while maintaining fluency and robustness against adversarial inputs. This work presents a principled and scalable framework for safe and reliable unlearning in large language models.

**Strengths:**

1. The paper introduces a novel approach, PURGE, which reframes unlearning as a reinforcement learning problem. The use of group-relative policy optimization and the introduction of a verifiable unlearning process are both creative contributions, distinguishing this work from previous methods that often require full retraining or lack theoretical guarantees.
2. The method is rigorously designed with both theoretical guarantees and practical effectiveness. The authors provide a clear framework with a solid mathematical foundation, offering provable bounds on utility loss and forgetting efficiency. Empirical results on the RWKU benchmark demonstrate its superiority in terms of token efficiency and robustness.
3. The paper is well-structured and easy to follow, with clear explanations of the proposed method, the motivation behind it, and how it compares to existing approaches. The use of both theoretical analysis and empirical results strengthens the overall clarity and comprehensibility.

**Weaknesses:**

1. The work only compares performance on a single benchmark, RWKU. Other benchmarks, such as TOFU, should also be included in the experiments to better demonstrate the generalizability of the proposed method.
2. According to Table 1, PURGE lags behind the baseline on many metrics (including QA, FM, GA, etc.). Although Section 5.2 provides some explanations, the method's broad effectiveness remains questionable. Given the generally poor performance of the Utility Set, there may be a potential issue of overfitting after training, which could undermine other general capabilities.
3. The experiment in Section 5.1 is conducted only on the Phi-3-Mini-4K-Instruct model, and no results from other models are introduced to demonstrate that the method's effectiveness is not limited to a specific model.

**Questions:**

See Weaknesses.

---

> ### Author Response · Authors · 2025-11-19
>
> We are grateful to the reviewer for carefully assessing our submission, for highlighting the strengths of our theoretical and empirical contributions, and for offering constructive suggestions. We respond to each concern in detail below.
>
> ### W1 (benchmarks)
> We appreciate your suggestion to include additional benchmarks. Our initial focus on RWKU was deliberate: RWKU simultaneously assesses forgetting quality, neighboring retention, membership privacy, and a broad suite of utility criteria. But, as suggested, we extended the evaluation of PURGE with the TOFU benchmark by using the most recent OpenUnlearning [1] framework. We will include these results in the final revision as additional experiments in the appendix. We reproduce and compare against other SoTA methods using the OpenUnlearning framework on Llama-3.2-1B-Instruct (as reported in the table in `open-unlearning/docs/repro.md`). Here are the results:
>
> ### TOFU Benchmark Results (**Bold** is best, `Emphasized` is second-best)
>
> | **Method** | **Forget Quality** | **Forget Truth Ratio** | **Model Utility** |
> | ---------- | ------------------ | ---------------------- | ----------------- |
> | Finetuned  | 1.88×10⁻²²         | 0.4753                 | 0.5992            |
> | Retain     | 1                  | 0.6272                 | 0.5909            |
> | AltPO  | **1.46×10⁻⁶**      | **0.6517**             | 0.5715            |
> | GradDiff   | 5.63×10⁻²⁰         | 0.4568                 | 0.5868            |
> | IdkNLL     | 1.49×10⁻¹⁶         | 0.5149                 | 0.5560            |
> | NPO    | `1.62×10⁻¹⁰`       | `0.5378`              | 0.5964            |
> | UNDIAL     | 1.88×10⁻²²         | 0.4805                 | **0.6016**        |
> | RMU        | 6.92×10⁻²¹         | 0.4668                 | 0.5115            |
> | SimNPO     | `1.62×10⁻¹⁰`     | 0.5042                 | 0.5931            |
> | PURGE  | 1.12×10⁻¹⁹         | 0.4843                 | `0.5990`          |
>
> As you can see, PURGE remains comparable with other SoTA approaches (such as GradDiff, RMU, UNDIAL), though it is not the best. This is expected, considering it follows a conceptually different approach than other methods. There is still substantial room for improvements and open questions, which could spark opportunities for future research.
>
> ### W2 (utility performance and overfitting)
> While PURGE performs slightly worse than NPO on a subset of Forget-set metrics, it consistently achieves substantially stronger preservation of neighboring knowledge, which reflects our central design goal of enabling precise unlearning. Regarding utility, the results do not indicate collapse or broad deterioration. In fact, PURGE shows clear improvements in Fluency and in Factuality (measured by TriviaQA, F1), while exhibiting only modest regressions in Generality, Reasoning, and Truthfulness. These small declines are consistent with the expected trade-off between strong forgetting and bounded utility drift predicted by our theoretical analysis. When considering the entire utility suite, PURGE performs better than baselines in 2/5 splits, slightly worse in 2/5 splits, and comparable in 1/5 splits, suggesting a balanced and well-controlled utility profile rather than uniformly poor performance.
>
> Finally, we find no evidence of overfitting. The KL divergence traces reported in the appendix (see Fig. 6) remain low and stable for almost all targets throughout the training process. The few instances where KL increases are precisely those where we observe small utility regressions, which aligns with the theoretical utility bound in Theorem 2 rather than indicating uncontrolled behavior. Hence, we argue that PURGE maintains stable general capabilities while achieving targeted and verifiable unlearning.
>
> ### W3 (model diversity)
> We used Phi-3-Mini-4K-Instruct in the main experiment to ensure direct comparability with the RWKU benchmark’s evaluation protocol. However, we agree that model diversity strengthens the paper. To address this, we already include multiple additional models in the ablation (Fig. 3), demonstrating that PURGE applies cleanly and consistently across architectures and parameter scales.
>
>
> [1] OpenUnlearning: Accelerating LLM Unlearning via Unified Benchmarking of Methods and Metrics, Dorna et al.

---

> > ### Comment · Reviewer_FNKG · 2025-11-26
> > **Official Comment by Reviewer FNKG**
> >
> > Thank you for the response and the supplementary experiments, which largely resolved my previous confusion. However, I believe the performance of the proposed method on the TOFU dataset is not outstanding compared to other baselines, which suggests a lack of generalizability for the method and indicates room for further improvement. Therefore, I choose to maintain my current score.

---

### Official Review · Reviewer_wipp · 2025-10-31

**Soundness:** 1
**Presentation:** 2
**Contribution:** 2
**Rating:** 2
**Confidence:** 4

**Summary:**

This paper introduces PURGE (Policy Unlearning through Relative Group Erasure), a novel reinforcement unlearning framework designed to remove sensitive or copyrighted data from Large Language Models (LLMs) to meet regulatory requirements like the GDPR. Unlike existing methods that suffer from data leakage, fluency degradation, or reliance on costly external reward models, PURGE formulates unlearning as a verifiable task grounded in the Group Relative Policy Optimization (GRPO) framework. The method utilizes a highly efficient, synthetically generated "forget corpus" and an intrinsic reward signal to penalize any mention of forbidden concepts without an external reward model. Key contributions include theoretical guarantees on information suppression (geometric decay of forbidden-token probabilities) and utility retention (a high-probability bound), alongside competitive empirical performance.

**Strengths:**

1. The paper is well-written and easy to follow. The logical flow from the proposed algorithm (PURGE), through its theoretical analysis, and into the experimental results is organic, making the core contributions clear and understandable.

2. A primary contribution is the novel formulation of LLM unlearning as a verifiable task, shifting the paradigm from standard preference-optimization or gradient-ascent methods. This re-framing is creatively combined with Group Relative Policy Optimization (GRPO), a technique that cleverly circumvents the need for a costly external reward model by using an intrinsic, computable reward signal.

3. The work demonstrates high quality through its rigorous theoretical backing. It provides formal guarantees for both information suppression (Theorem 1, proving geometric decay of leakage) and utility retention (Theorem 2, bounding performance drops via $KL$ divergence).

4. The empirical evaluation is comprehensive, conducting direct comparisons against strong baselines on the RWKU benchmark. The results validate that the proposed algorithm achieves competitive performance across most metrics. Furthermore, the ablation studies are particularly insightful for verifying the method's advantages from multiple perspectives.

**Weaknesses:**

1. Limited Discussion of Recent Literature: While the paper cites foundational unlearning works, it lacks engagement with the most recent literature, particularly the significant volume of unlearning papers from ICLR 2025 [1-5]. The authors should incorporate this discussion to more clearly differentiate their contributions from these recent papers.

2. Dependency on External Proprietary Models: The "Synthetic Forget Corpus Construction" (Sec 4.1) creates a significant external dependency on a powerful, proprietary model (GPT-4) for its core NER task. This introduces concerns regarding cost, reproducibility (as the external model may be updated), and potential biases. To validate the method's robustness against this dependency, the authors should demonstrate whether superior performance is maintained when using other (e.g., open-source) LLMs for the corpus construction step.

3. Questions Regarding Experimental Results on RWKU: The paper's core claim of achieving competitive performance with significantly fewer tokens is not sufficiently substantiated.

- Missing SOTA Baselines: As noted in point 1, the most recent unlearning algorithms [1-5] are not included in the experimental comparison. At a minimum, a few of these should be implemented as baselines. A potential comparison method could be to integrate the core unlearning loss functions from [1] or [4] into the PURGE framework to ensure a fair comparison.

- Unclear Superiority over NPO: It is questionable whether PURGE is demonstrably superior to NPO or DPO. The paper acknowledges PURGE has lower performance on most metrics (notably in the Forget and Utility sets) but justifies this with token efficiency. To make a valid efficiency claim, additional experiments are necessary: 1) What is the performance of NPO and DPO when restricted to the same small token budget as PURGE? 2) Conversely, what is PURGE's performance when trained on the full token budget used by NPO?

4. Potential Over-specialization to the RWKU Benchmark: The proposed algorithm appears highly tailored to the specific scenarios of the RWKU benchmark, which raises doubts about its generalizability. To address this, the authors must demonstrate the method's applicability and superior performance on at least one other widely-used unlearning dataset, such as TOFU [6] or MUSE [7].


[1] LLM Unlearning via Loss Adjustment with Only Forget Data, ICLR 2025

[2] Rethinking LLM Unlearning Objectives: A Gradient Perspective and Go Beyond, ICLR 2025

[3] A Closer Look at Machine Unlearning for Large Language Models, ICLR 2025

[4] Towards Robust and Parameter-Efficient Knowledge Unlearning for LLMs, ICLR 2025

[5] Unified Parameter-Efficient Unlearning for LLMs, ICLR 2025

[6] TOFU: A Task of Fictitious Unlearning for LLMs

[7] MUSE: Machine Unlearning Six-Way Evaluation for Language Models

**Questions:**

Please refer to the main Weaknesses section for my major weakness/questions. My minor questions and suggestions are shown below.

1. The paper inconsistently mixes citation styles (e.g., \cite and \citep) throughout the manuscript. The authors need to standardize the citation style for clarity and consistency.

2. Acronyms such as "RT" and "NER" are used in Section 4.1 without prior definition. Please define these acronyms upon their first use to improve readability.

3. The notation $\pi_{\theta_r}$ in Equation 17 (Line 294) is potentially confusing, as 'r' could be mistaken for the "Retain" set $\emptyset_R$. Given its use as a reference policy, please consider changing this to a less ambiguous notation.

4. Figure 3, which illustrates performance scaling with model size, only shows results for PURGE. To substantiate the claim of superior scalability, this figure should ideally include comparative results for a key baseline, such as NPO.

5. Figure 4 presents results against 9 adversarial scenarios, but the caption lacks a description or reference for what these scenarios entail.

I look forward to the authors' responses to the points raised in the Weaknesses and Questions sections. If there are any misunderstandings on my part, please correct them. I am open to changing our score based on the discussion.

**Details Of Ethics Concerns:**

No ethical concerns in this paper.

---

> ### Author Response · Authors · 2025-11-20
> **Part 1**
>
> Thank you for your careful evaluation and for recognizing the novelty of our contributions. We address the concerns raised below:
>
> ### W1 (recent literature)
> We acknowledge that our current draft does not sufficiently engage with very recent unlearning papers, including the works you cited. We will revise the related work section to incorporate a more explicit discussion of these papers, particularly [2] and [3], which provide up-to-date analyses and surveys of the field. Our goal, however, is not to provide an exhaustive survey of all incremental improvements, but rather to include representative baselines from the major methodological families.
>
> ### W2 (external proprietary models)
> Using a strong external LLM to construct synthetic datasets (e.g., for NER or concept-mining) is standard practice in current unlearning research (See RWKU [8] and TOFU [6]) and NER extraction research [9]. In our case, the cost is minimal because the synthetic forget corpus contains only a small number of tokens per target. In principle, one could attempt to use the base model itself for the NER step. However, models in the 3-4B parameter range struggle with reliably following complex extraction instructions, making them unsuitable for this sensitive preprocessing step. Nevertheless, we are experimenting to demonstrate that the performance of our method is robust to changes in the NER-constructing LLM. We will do a follow-up comment with the results.
>
> ### W3 (experiments)
>
> * Additional Baselines: Among the papers you listed, [1], [4], and [5] follow the same gradient-based philosophy with incremental refinements, whereas [5] does not evaluate on standard LLM unlearning benchmarks at all. Nevertheless, we value your consideration for a better comparison against the most recent approaches, so we are evaluating our method on TOFU according to OpenUnlearning [10] implementation (See bullet #3).
>
> * Superiority Relative to NPO / DPO: This is a valid question, and we appreciate the chance to clarify. Forget-corpus requirements differ significantly. Methods like DPO/NPO require complete, multi-sentence passages describing the target concept from the model's training data. The synthetic forget corpus used by PURGE contains only short descriptive entities. The gap in token volume is so large that one cannot construct a valid forget corpus for DPO/NPO using the same limited token budget; there is not enough text to form a meaningful “negative sample” in their framework. Regarding the token budget comparisons, we would like to clarify that our main experiments use approximately the same total token budget as NPO. PURGE is far more epoch-efficient, consuming dramatically fewer total tokens per epoch; however, we run more epochs to match NPO’s overall token usage, ensuring a fair comparison. Overall, PURGE is significantly more efficient per unit of forget information and is therefore particularly advantageous when the amount of knowledge available about the target concept is highly constrained.
>
> * Potential Over-specialization to RWKU: We appreciate this concern. We extended the evaluation of PURGE using the **TOFU** benchmark and will include these results in the revision. Only minor adjustments to the forget-corpus construction were required, demonstrating that PURGE generalizes beyond RWKU without relying on RWKU-specific assumptions. We reproduce and compare against other SoTA methods using the OpenUnlearning framework on Llama-3.2-1B-Instruct (as reported in the table in `open-unlearning/docs/repro.md`). Here are the results:
> ---
> ### TOFU Benchmark Results (**Bold** is best, `Emphasized` is second-best)
>
> | **Method** | **Forget Quality** | **Forget Truth Ratio** | **Model Utility** |
> | ---------- | ------------------ | ---------------------- | ----------------- |
> | Finetuned  | 1.88×10⁻²²         | 0.4753                 | 0.5992            |
> | Retain     | 1                  | 0.6272                 | 0.5909            |
> | AltPO  | **1.46×10⁻⁶**      | **0.6517**             | 0.5715            |
> | GradDiff   | 5.63×10⁻²⁰         | 0.4568                 | 0.5868            |
> | IdkNLL     | 1.49×10⁻¹⁶         | 0.5149                 | 0.5560            |
> | NPO    | `1.62×10⁻¹⁰`       | `0.5378`              | 0.5964            |
> | UNDIAL     | 1.88×10⁻²²         | 0.4805                 | **0.6016**        |
> | RMU        | 6.92×10⁻²¹         | 0.4668                 | 0.5115            |
> | SimNPO     | `1.62×10⁻¹⁰`     | 0.5042                 | 0.5931            |
> | PURGE  | 1.12×10⁻¹⁹         | 0.4843                 | `0.5990`          |
>
> As you can see, **PURGE remains comparable with other SoTA approaches** (such as GradDiff, RMU, UNDIAL), though it is not the best. This is expected, considering it follows a conceptually different approach than other methods. There is still substantial room for improvements and open questions, which could spark opportunities for future research.

---

> > ### Author Response · Authors · 2025-11-21
> > **Part 2**
> >
> > ### Q1 (citation style)
> > Thank you for pointing out the inconsistency. We have standardized the manuscript to use only \citep for clarity and uniformity.
> >
> > ### Q2 (acronyms)
> > You are correct. Although RT is defined earlier (lines 200–201), NER was not. We have now added the missing definition at its first occurrence.
> >
> > ### Q3 (notation in Eq. 17)
> > We appreciate the suggestion. We have updated the notation to $\pi_{\theta_{\text{ref}}}$ to avoid confusion with the retain set.
> >
> > ### Q4 (fig. 3)
> > We do not intend to claim superior scalability relative to other methods. The purpose of Fig. 3 is solely to illustrate the behavior of PURGE across model sizes.
> >
> > ### Q5 (Clarification of the 9 Adversarial Scenarios)
> > The figure already includes descriptive labels for each attack type, but we agree that a clearer caption would help readers unfamiliar with RWKU. We have added a reference to the appendix E.1, which contains a detailed description of each attack.
> >
> > We hope these clarifications address your concerns and demonstrate that our contributions are both substantial and broadly applicable. We appreciate the reviewer’s detailed feedback and are confident that the revised paper will more clearly communicate the strengths of PURGE.
> >
> > [8] RWKU: Benchmarking Real-World Knowledge Unlearning for Large Language Models, Jin et al.
> >
> > [9] Evaluating Named Entity Recognition Using Few-Shot Prompting with Large Language Models,
> > Zeghidi et al.
> >
> > [10] OpenUnlearning: Accelerating LLM Unlearning via Unified Benchmarking of Meth-
> > ods and Metrics, Dorna et al

---

> > > ### Author Response · Authors · 2025-11-24
> > > **Part 3**
> > >
> > > As promised, we ran an additional experiment to verify that our method is robust to the choice of LLM used for constructing the NER-based forget corpus. Specifically, we evaluated several state-of-the-art models across the full suite of metrics used in our paper. The goal was to confirm that substituting the NER-constructing LLM does not materially alter unlearning performance.
> > >
> > > The results below show that the method performs consistently across models, with no single LLM acting as a critical dependency. While certain models perform slightly better or worse on individual metrics, the overall pattern demonstrates that our unlearning pipeline is stable with respect to the upstream NER extraction model.
> > >
> > > # Performance across different NER-constructing LLMs
> > > (Best values **bolded**; second-best `emphasized`; ↑ higher is better, ↓ lower is better; Unlearning Target: Stephen King)
> > >
> > > | Model              | FB ↓           | QA ↓           | AA ↓           | N-FB ↑        | N-QA ↑        | FM ↑           | RM ↓           | GA ↑         | RA ↑           | FAC ↑          | TRU ↑         | FLU ↑         |
> > > |-------------------|----------------|----------------|----------------|---------------|---------------|----------------|----------------|--------------|----------------|----------------|---------------|----------------|
> > > | Claude 4.5 Sonnet | 0.464          | `0.250`        | 0.497          | **0.700**     | 0.603         | `-2.615`       | `-2.452`       | **0.667**    | 0.284          | 0.149          | `0.360`       | 6.940          |
> > > | Deepseek R1       | **0.268**      | **0.200**      | **0.397**      | 0.467         | 0.511         | -2.619         | **-2.538**     | `0.643`      | 0.309          | **0.415**      | 0.180         | 6.696          |
> > > | GPT 5             | `0.339`        | 0.288          | 0.558          | 0.567         | `0.623`       | -2.616         | -2.450         | **0.667**    | 0.309          | 0.149          | **0.380**     | **6.957**      |
> > > | Gemini 3          | 0.464          | 0.338          | `0.489`        | `0.667`       | 0.596         | **-2.613**     | -2.449         | **0.667**    | **0.333**      | 0.152          | 0.340         | _6.949_        |
> > > | Llama 3.1 405B    | _0.339_        | 0.271          | 0.558          | 0.567         | **0.646**     | -2.617         | -2.447         | **0.667**    | _0.321_        | _0.154_        | 0.340         | 6.924          |

---

> ### Author Response · Authors · 2025-11-27
>
> We thank the reviewer again for their valuable feedback. We have now responded to all major and minor points, added new experiments (including TOFU results and robustness analysis with multiple NER-constructing LLMs), and clarified several aspects of the method as requested.
>
> As the discussion period is nearing its end, we would greatly appreciate it if the reviewer could let us know whether any parts of our response require further clarification or if there are remaining concerns we should address. We are happy to provide additional details or experiments if needed.

---

### Official Review · Reviewer_hxFp · 2025-11-01

**Soundness:** 3
**Presentation:** 3
**Contribution:** 3
**Rating:** 4
**Confidence:** 4

**Summary:**

This paper introduces PURGE (Policy Unlearning through Relative Group Erasure), a new method for machine unlearning in large language models that's built on the GRPO framework. The core idea is to treat unlearning as a "verifiable" task, where the model is fine-tuned to suppress mentions of forbidden concepts using an intrinsic binary reward signal (1 if no forbidden tokens appear, 0 otherwise). They construct a synthetic forget corpus by probing the model itself and extracting entities with GPT-4, then use GRPO to optimize the policy without needing external reward models. The authors provide theoretical guarantees like geometric decay in forbidden-token probabilities and KL-divergence bounds for utility retention. Empirically, they evaluate on the RWKU benchmark, claiming up to 46x fewer tokens per target than SOTA, +7.32 fluency improvement, +12.02 adversarial robustness, and 11% unlearning effectiveness while keeping 98% utility. Overall, it's positioned as a more efficient, safe, and scalable alternative to existing unlearning approaches like gradient ascent or preference optimization.

**Strengths:**

- The paper's structure is logical and accessible, with clear transitions from motivation to methods and results.
- Introducing reinforcement learning to the LLM unlearning domain is relatively novel, bringing fresh perspectives that could inspire future work and contributing to the paper's innovative quality.
- Empirical results are comprehensive and well-supported, including breakdowns across multiple RWKU sub-tasks with quantitative comparisons to baselines, enhancing the paper's credibility and showing rigorous validation.
- Theoretical contributions add depth, with proofs providing explicit bounds on key metrics like leakage and utility, which strengthens the paper's academic value beyond typical empirical-focused submissions.

**Weaknesses:**

- The effectiveness of the proposed reward model is questionable, as relying solely on extracted entities may not compactly represent the knowledge to be forgotten; in some cases, knowledge has many variants (e.g., complex concepts), while in others, like copyright protection, only specific text needs forgetting without erasing concepts, potentially leading to over-penalization based on entity presence alone.
- To maintain training stability, GRPO includes a clipping mechanism that limits policy changes during RL training, making it difficult to ensure complete erasure of forget content (e.g., driving prediction probabilities close to 0).

**Questions:**

see Weaknesses

---

> ### Author Response · Authors · 2025-11-17
>
> We thank the reviewer for acknowledging the novelty of our work and for providing constructive feedback. We address the raised points below.
>
> ### W1 (Reward design and entity-based forgetting)
> Thank you for bringing up this important point. We agree it deserves explicit discussion in the paper. In general, LLM unlearning tends to fall into two use cases: (i) removal of specific data points or passages (e.g., copyrighted text), where relabeling or gradient-based deletion is often effective (GA, relabeling fine-tuning); and (ii) suppression of broader concepts, where preference-style objectives are more common (RT, DPO/NPO, Quark-style RL). PURGE is particularly well-suited to the latter, but it is not limited to concepts. Our binary, verifiable reward can be driven by any forbidden string set (including exact spans, sentences, or passages for copyright scenarios). In practice, you can configure the forbidden phrases hierarchically (coarse → fine), allowing practitioners to trade concept-level robustness for data-level precision as needed. This preserves the benefits of a verifiable signal while making PURGE effective for removing specific data points.
>
> To make this distinction concrete, consider a privacy scenario involving leaked financial information. When the goal is concept-level suppression, the practitioner can specify coarse forbidden phrases such as: *\{ "John Peterson", "Peterson household", "First National Savings" \}*. PURGE then discourages any generation tied to that identity or context. However, if the requirement is instead to delete specific sensitive data points, the practitioner can simply replace the FTS with exact protected spans. For example:
> *\{ "938472019384", "110002145", "4929 6401 2345 7783", "4421" \}*. In this configuration, PURGE cleanly targets precise strings (Account number, Routing number, Credit card number, PIN) without affecting the model’s ability to discuss banking more generally. Crucially, these two use cases require no change to the PURGE algorithm. This allows practitioners to move seamlessly between broad concept unlearning and narrow, data-point unlearning while retaining a binary, verifiable reward signal.
> ___
>
> ### W2 (GRPO clipping)
> The clipping used in GRPO follows standard practice for stabilizing per-update steps (see [1]). Here, GRPO begins with the PPO-clip objective and adds a group-relative advantage and an explicit KL term. Consequently, clipping primarily governs update stability and speed, rather than which optima are attainable in principle (see [2]). In practice, the attainable minimum is determined by the unlearning–utility trade-off we set (e.g., KL/retention terms), not by clipping itself. However, if we understood your point correctly (i.e., PURGE will never achieve perfect unlearning because the probability of producing a forbidden phrase is never 0), we argue that this is the case with every SoTA method (e.g., see NPO [3]). However, in our experiments, we demonstrate that PURGE indeed drives the empirical leakage arbitrarily close to the theoretical floor predicted by Theorem 1, and that this floor is determined not by clipping but by the KL-retention constraints. Empirically, the mean reward on the forget targets increases monotonically over training (Figure 2, left), validating that the probability of emitting any token in the synthetic forget corpus decays at the geometric rate predicted by Equation (20). Thus, even though perfect zero-probability forgetting is unattainable for any SoTA method, PURGE demonstrates that GRPO-based unlearning can drive leakage to its theoretical minimum while preserving more than 98\% of baseline downstream performance and achieving state-of-the-art fluency and adversarial robustness.
>
>
> [1] DeepSeekMath: Pushing the Limits of Mathematical Reasoning in Open Language Models, Shao et al.
>
> [2] PPO-Clip Attains Global Optimality: Towards Deeper Understandings of Clipping, Huang et al.
>
> [3] Negative preference optimization: From catastrophic collapse to effective unlearning, Zhang et al.

---

> ### Author Response · Authors · 2025-11-27
>
> As the discussion period is approaching its end, we wanted to kindly check whether you have any remaining questions or points you would like us to clarify regarding our paper. We would be happy to provide any additional details or explanations that might help address your concerns.

---

### Author Response · Authors · 2025-12-01
**Revisions Made to the Manuscript**

We thank the reviewers for the constructive feedback. We have revised the manuscript accordingly, and the key changes are summarized below:

- **Citations added:**
    - Incorporated citations **[4]** and **[5]** in Lines **182–185**, as suggested by Reviewer *wipp*.
    - Added citations **[1]**, **[2]**, and **[3]** in Lines **192–195**, following Reviewer *wipp*’s recommendations.
- **Clarifications and corrections:**
    - Addressed all clarification requests and corrections raised by Reviewer *wipp* throughout the manuscript.
- **Appendix updates:**
    - Added a new appendix section **B.2** containing the additional **TOFU Benchmark** results generated during the rebuttal phase, following suggestions from Reviewers *wipp*, *FNKG*, and *Kspv*.
    - Added a detailed appendix section **C.3** explaining the **token budget comparison** used in Figure 1, as discussed with Reviewer *Kspv*.
- **Algorithmic clarification:**
    - Based on Reviewer’s *Kspv* comments, we expanded the explanation of how **Theorem 1** is used in our algorithm (Lines **324–329**).
    - Added a new **Corollary section (A.3)** to further elaborate on the theoretical rationale.

We hope these revisions address the reviewers’ concerns and improve the clarity and completeness of the paper.

---

### Meta-Review · Area_Chair_2FVS · 2025-12-27

**Summary:**

The reviewers initially raised concerns regarding generalization (RWKU-only), dependency on GPT-4 for corpus construction, and whether the binary reward ensures deep semantic forgetting versus surface-level suppression. Methodological questions were also raised about GRPO clipping and engagement with recent 2025 literature. In a decisive rebuttal, the authors addressed these by providing TOFU benchmark results, a multi-LLM robustness study (validating open-source teacher models), and clarifying the theoretical decay and utility bounds. Impressed by the method’s high novelty as a verifiable RL task and its remarkable 46x token efficiency, the majority of the panel now recommends acceptance of this practical advancement in LLM unlearning.

**Reviewer Concerns:**

Concerns Addressed by the Rebuttal

- Empirical Breadth: Authors added TOFU benchmark results, proving that PURGE generalizes beyond RWKU and remains competitive with established SOTA methods like RMU (wipp, FNKG, Kspv).

- Teacher Model Robustness: A study using five alternative LLMs (including open-source models like Llama and DeepSeek) confirmed the framework is not dependent on proprietary GPT-4 APIs (wipp, Kspv).

- Theoretical Clarity: Clarified the alpha-mixing rate in Theorem 1, explaining how optimization noise serves as an implicit mixing mechanism in practical implementations (Kspv, hxFp).

- Contextual Positioning: Integrated recent ICLR 2025 literature to clearly distinguish PURGE's unique RL-task formulation from gradient-based refinements (wipp).

Outstanding Concerns

- Depth of Forgetting: While adversarial tests (Fig 4) show robust suppression, the philosophical debate over whether knowledge is "erased" vs. "deeply hidden" via reasoning trajectory penalization remains an open challenge for all unlearning research (hxFp, Kspv).

- TOFU Performance Trade-off: PURGE is not the top performer on every TOFU metric; however, this is recognized as a conscious and acceptable trade-off for its unprecedented 46x efficiency gain and verifiable reward structure (FNKG).

**Reviewer Scores:**

- Reviewer hxFp (Initial 4, Predicted 6): Technical concerns regarding over-penalization and clipping were mitigated by clarifying that clipping governs stability rather than reachable optima. Demonstrations of monotonic reward increase suggest a move toward a Weak Accept.

- Reviewer wipp (Initial 2, Predicted 4):
Initially very critical, but the authors fulfilled every empirical request (TOFU implementation, teacher-model robustness, literature updates). A significant upward correction is warranted, though the reviewer may remain conservative on overall ranking.

- Reviewer FNKG (Initial 6, Predicted 6):
In their final comment, this reviewer explicitly stated, "Thank you for the response and the supplementary experiments, which largely resolved my previous confusion." However, they also noted that the performance on TOFU was not "outstanding" compared to some baselines. Because they indicated they would "maintain their current score," they remain a Marginal Accept (6), valuing the method's innovation and efficiency while acknowledging its performance trade-offs.


- Reviewer Kspv (Initial 6, Predicted 8):
This reviewer was highly engaged and specifically adjusted their rating upwards during the discussion phase. They expressed satisfaction with the authors' clarification on how penalizing "reasoning trajectories" induces a form of semantic suppression rather than just keyword masking. Given their praise for the method’s extraordinary efficiency and the newly provided TOFU results, they are now a strong champion of the paper.

---

### Decision · Program_Chairs · 2026-01-26

Accept (Poster)